# How effective and cost-effective are innovative combinatorial technologies and practices for supporting older people with long-term conditions to remain well in the community? An evaluation protocol for an NHS Test Bed in North West England

Sandra Varey,[1] Alejandra Hernández,[1] Tom M Palmer,[2] Céu Mateus,[3] Joann Wilkinson,[1] Mandy Dixon,[4] Christine Milligan[1]

¹Centre for Ageing Research, Division of Health Research, Lancaster University, Lancaster, UK
²Department of Mathematics and Statistics, Lancaster University, Lancaster, UK
³Health Economics, Division of Health Research, Lancaster University, Lancaster, UK
⁴Lancaster Health Hub, Lancaster University, Lancaster, UK

**Correspondence to**
Professor Christine Milligan;
c.milligan@lancaster.ac.uk

## ABSTRACT

**Introduction** The Lancashire and Cumbria Innovation Alliance (LCIA) Test Bed is a partnership between the National Health Service in England, industry (led by Philips) and Lancaster University. Through the implementation of a combination of innovative health technologies and practices, it aims to determine the most effective and cost-effective ways of supporting frail older people with long-term conditions to remain well in the community. Among the Test Bed's objectives are to improve patient activation and the ability of older people to self-care at home, reduce healthcare system utilisation, and deliver increased workforce productivity.

**Methods and analysis** Patients aged 55 years and over are recruited to four cohorts defined by their risk of hospital admission, with long-term conditions including chronic obstructive pulmonary disease, dementia, diabetes and heart failure. The programme is determined on an individual basis, with a range of technologies available. The evaluation is adopting a two-phase approach: phase 1 includes a bespoke patient survey and a mass matched control analysis; and phase 2 is using observational interviews with patients, and weekly diaries, action learning meetings and focus groups with members of staff and other key stakeholders. Phase 1 data analysis consists of a statistical evaluation of the effectiveness of the programme. A health economic analysis of its costs and associated cost changes will be undertaken. Phase 2 data will be analysed thematically with the aid of Atlas. ti qualitative software. The evaluation is located within a logic model framework, to consider the processes, management and participation that may have implications for the Test Bed's success.

**Ethics and dissemination** The LCIA Test Bed evaluation has received ethical approval from the Health Research Authority and Lancaster University's Faculty of Health and Medicine Research Ethics Committee. A range of dissemination methods are adopted, including deliberative panels to validate findings and develop outcomes for policy and practice.

## Strengths and limitations of this study

► The evaluation is considering both the effectiveness, in terms of measures such as hospital admissions, and cost-effectiveness of the Test Bed.
► The protocol evaluates the potential for patient activation through health technology.
► Observational interviews with patients in their own homes enable insights into how older people experience and engage with the technology.
► For both the Test Bed participants and controls, the evaluation has to use proxy measures of the costs incurred to the National Health Service.
► Due to the timescale of the overall programme, it is not possible to assess the longer term impact of the programme.

## INTRODUCTION

In the UK, increasing numbers of older people are living with long-term conditions (LTCs) including diabetes, chronic obstructive pulmonary disease (COPD), heart failure and dementia.[1] This is contributing to increased pressure on health and social care systems, including accident and emergency services, which in turn increases demand on the number of hospital beds.[2] At the same time, local authorities are experiencing a reduction in the funding available to provide social care.[3]

Concerns about this so-called care gap and our future ability to cope with the growing numbers of older people with LTCs underpin governmental drivers to develop

**Table 1** Recruitment criteria for the four Lancashire and Cumbria Innovation Alliance Test Bed cohorts

| Cohort | Age | Risk of hospital admission (%) | Long-term conditions |
|---|---|---|---|
| Cohort 1 | Aged 55 years or over | >25 | Chronic obstructive pulmonary disease, heart failure |
| Cohort 2 | | >10 and <25 | |
| Cohort 3 | | <10 | Diabetes, asthma, chronic heart disease, hypertension |
| Cohort 4 | | NA | Early-stage dementia (Addenbrooke's Cognitive Examination-III assessment tool) |

NA, not applicable.

new models of care within vanguard sites[i] that are cost-effective and that facilitate older people's ability to better self-manage their care needs at home. The NHS England Test Bed programme has thus been designed to encourage the trialling of new models of care that are supported by health technologies across a number of areas in the UK.

This paper reports on the protocol for the evaluation of one such programme, the Lancashire and Cumbria Innovation Alliance (LCIA) Test Bed in the North West of England. The programme commenced on 1 April 2016 and is due to be completed on 30 June 2018.

### The LCIA Test Bed initiative

The LCIA Test Bed[ii] is a partnership between the National Health Service (NHS), industry (led by Philips) and Lancaster University, and is one of seven Test Beds located across England. Two neighbouring vanguard sites, part of the Lancaster Health Hub (an established NHS/university partnership comprising 10 local organisations), will deliver the Test Bed. The Test Bed's aims and objectives are presented below.

### Test Bed aim

To determine the most effective and cost-effective ways of supporting frail older people with dementia and other LTCs to remain well in the community and avoid unnecessary hospital admissions.

### Test Bed objectives

To use a combinatorial range of technologies and services to:
► better support frail older people, living with LTCs
► improve patient activation[iii] and the ability of older people with a range of LTCs to self-care at home

► improve health awareness and outcomes for older people with LTCs
► reduce healthcare system utilisation and increase productivity within the healthcare workforce.

Over 2 years, the Test Bed is implementing a combination of innovative technologies and practices aimed at supporting these aims and objectives. The clinical challenge is to reduce hospital admissions, create capacity, reduce overall costs, minimise other health and social care usage, and maximise patient outcomes for a targeted population of older people (defined in this Test Bed as those aged 55 years and over) who suffer a range of LTCs including dementia. These LTCs present a major challenge for the area given its dispersed population, and innovative solutions are urgently required.

The main technology partner is Philips, which together with a number of small-sized and medium-sized enterprises, social enterprises and voluntary organisations is working with the Test Bed to introduce a health technology-enabled supported self-care programme.[iv]

The two vanguard sites include the Fylde Coast and North Lancashire, with the technologies being provided in addition to the new models of care being implemented by these two sites. The Fylde Coast Vanguard[v] is drawing on the extensive care model, which brings together a range of services in the same location to achieve a coordinated team of healthcare professionals, with patients each allocated a well-being support worker.[4] This model focuses on multispecialty community providers, moving specialist care out of hospitals and into the community.[5] The North Lancashire Vanguard, Better Care Together, sees healthcare professionals working together in partnership to help people manage their own health conditions.[6] This model focuses on integrated primary and acute care settings, joining up general practitioner (GP), hospital, community and mental health services.[7] Further details of these new models of care are set out in the NHS England 'Five Year Forward View'[8] and by the Nuffield Trust.[9]

---

[i]In 2015, the National Health Service in England selected 50 vanguard sites to take a lead on the development of new integrated care models designed to act as blueprints for the future of the NHS. Complete redesign of whole health and care systems is being considered. The hope is that the vanguards will provide inspiration to the rest of the health and care system across the UK.[14] For more information about NHS vanguards, visit https://www.england.nhs.uk/ourwork/new-care-models/vanguards/.

[ii]For more information about the LCIA Test Bed, visit https://www.england.nhs.uk/ourwork/innovation/test-beds/innovation-alliance/.

[iii]For more information about the NHS definition of patient activation, visit https://www.england.nhs.uk/ourwork/patient-participation/

self-care/patient-activation/pa-faqs/.

[iv]Cambridge Cognition Limited, Good Things Foundation, Intelesant Limited, MKS Solutions Limited, Simple Shared Healthcare Limited and uMotif Limited.

[v]The Fylde Coast Vanguard encompasses NHS Blackpool CCG, NHS Fylde & Wyre CCG and Blackpool Teaching Hospitals.

As part of the LCIA Test Bed programme, these existing services are being combined with innovative technologies and support for older people with LTCs including COPD, heart failure, diabetes and dementia. For those with lower levels of risk in cohort 3 (see table 1), asthma and hypertension will also be included. The focus of this Test Bed is to empower patients to actively manage their health conditions and change behaviour through the provision of appropriate technologies, coaching and education.

A core part of the programme also focuses on increasing involvement of the third sector and community organisations in providing support for the Test Bed model, and understanding the impact of the service on staff and key stakeholders.

### Core outcomes for the evaluation

The Test Bed focuses on a combinational approach to technologies. Hence, rather than focusing on the impact of any single technology, the Test Bed evaluation is focusing on how health technologies as an addition to existing vanguard services can improve and promote self-care at home and reduce hospital admissions for older people with LTCs.

The evaluation is focusing on three primary outcomes:
1. the extent to which supported self-care telehealth technology can improve patient outcomes, through reduced hospital admissions, medication, GP visits, home visits and community service use
2. the extent to which supported self-care telehealth technology can improve patient activation for frail older people with LTCs
3. the cost-effectiveness of the programme and how it might be scaled up to provide better value for patients and taxpayers.

Secondary outcomes include the following:
1. identification of the strengths and weaknesses of the programme and how these may be built on or addressed
2. assessment of patient/staff perceptions of how the new service improves on existing services and makes best use of voluntary and community services
3. assessment of the impact of the new service within the healthcare workforce in terms of communication between care teams, productivity, capacity, coordination of care and work satisfaction.

To achieve these, we are:
1. using quantitative and qualitative measures to examine benefits to patients and patient outcomes as identified above
2. undertaking an economic evaluation of the impact of adding a combination of technologies to support and promote self-care in the provision of healthcare in comparison with standard care with cohorts 1 and 2 (see table 2)
3. drawing on a logic model to demonstrate how to construct the service as a scalable model that might provide better value to patients and taxpayers.

## METHODS AND ANALYSIS
### Ethical approval

The overall evaluation involves two distinct phases: phase 1 involves collection of survey data designed to evaluate any change in health and health-related behaviours in patients using the combinatorial technologies; and phase 2 involves understanding the impact and experiences of patients using the technologies, and the impact

**Table 2** Lancashire and Cumbria Innovation Alliance Test Bed cohort sample sizes

| Cohort | Total population (%)* | Total patients | Frail and elderly patients (%) | Patients with dementia (%) | Patients estimated to be appropriate for the service (n) | Appropriate patients within cohort that will be recruited within the Test Bed period (%) | Minimum number of patients recruited within Test Bed period | Estimated number of patients by the end of the Test Bed period |
|---|---|---|---|---|---|---|---|---|
| (Age >55, risk >25% and 1 LTC) | 1.70 | 5262 | 20 | 3.5 | 1737 | 30 | 521 | 500 |
| (Age >55, 10%< risk <25% and 1 LTC) | 3.00 | 9286 | 4 | 1.1 | 3064 | 15 | 460 | 500 |
| (Age >55, risk <10% 1 of a broader set of LTCs) | 9.20 | 28 477 | 2 | 1.6 | 9398 | 5 | 470 | 500 |
| (Diagnosis of dementia) | 0.79 | 2430 | | 100.0 | 810 | 10 | 81 | 100 |

(*Combined over a population of approximately 310 000 people) (see Acknowledgements).
LTC, long-term condition.

on working practices among members of staff involved in the delivery of the technologies. Following discussion with the Health Research Authority (HRA) and the Research and Development Directors of the NHS Trusts involved in the programme, it was agreed that phase 1 was service evaluation and as such did not require HRA approvals. Phase 2 however was deemed by the HRA to be gathering new data and was thus defined by them as 'research', therefore requiring HRA approvals (Integrated Research Application System (IRAS) Project ID: 208395). Further details about the two phases are provided below.

Phase 2 of the evaluation has also been adopted onto the National Institute for Health Research portfolio.[vi]

### Sampling and recruitment

Patients are being recruited to four cohorts defined by their risk of hospital admission. The LTCs are noted above and are outlined in table 1.

Participants for cohorts 1, 2 and 3 are being recruited to the Test Bed through the vanguard services to which they are already assigned. As illustrated in table 1, cohort 4 is focusing specifically on patients with early-stage dementia. Potential participants for cohort 4 are being recruited through the Memory Assessment Service (MAS) in the Fylde Coast and North Lancashire. In the diagnosis of dementia, these services use the Addenbrooke's Cognitive Examination-III (ACE-III) assessment tool to focus on five cognitive domains. People who have received a recent diagnosis of mild dementia by a consultant psychiatrist following their comprehensive assessment with the MAS are deemed as having fulfilled the entry criteria of mild dementia. For these patients, the ACE-III or Mini-ACE has been completed as part of their cognitive assessment at the MAS clinic, the results of which have aided the formulation of the dementia diagnosis. Only patients judged by a clinician to have mild dementia are invited to take part in the Test Bed.

All patients consenting to take part in the Test Bed and receiving the technology are required to participate in some aspect of phase 1 of the service evaluation and will sign a consent form. Specifically, primary outcomes 1, 2 and 3 will be evaluated for cohorts 1 and 2, and primary outcomes 1 and 2 will be evaluated for cohorts 3 and 4. Participation in phase 2, the qualitative phase of the evaluation, is optional. Patients indicating an interest in participating in phase 2 are being consented separately through a two-stage consent process.

The sample size for recruitment to each cohort has been informed by population and risk data for the area, along with the methodology used in a previous study.[10 11] As table 2 illustrates, the overall aim of the Test Bed is to recruit 1600 patients; however, given the anticipated challenges of recruitment and potential attrition

due to the nature of the patient group, we estimate an overall recruitment and retention figure of 60% (overall n=960).

Allowing for a phased increase in recruitment, the overall recruitment is estimated at approximately 60%–70% of the figures identified in table 2. This will provide sufficient data from cohorts 1 and 2 to demonstrate a change in hospital admissions (a primary outcome). Cohorts 3 and 4 will not be assessed for this outcome, although other measures of well-being, patient activation and loneliness will be used to describe these cohorts. The secondary outcomes are being supported by observational interviews across all four cohorts.

### The programme

The patient interface varies across the four cohorts, and the technologies available to people within the different cohorts are presented in table 3. A patient's programme is determined on an individual basis and is informed by a number of factors, including health condition(s), personal preferences and other needs. Some patients are using different technologies at different periods of time within the 6-month programme.

In summary, the LCIA Test Bed provides a structured programme of case management, monitoring, education and coaching supported by a clinical hub. It thus involves clinical and non-clinical teams, community involvement and community feedback. Good Things Foundation[vii] is providing support to patients regarding their digital skills to help them use the innovator technologies.

### Evaluation design

A requirement of this evaluation[viii] was that it adopts a rapid cycle approach to enable the delivery of the programme to be adapted and amended in the light of feedback received from patients and clinicians. As a result, a randomised controlled trial design was deemed unsuitable.

To address the core aims and objectives outlined above and the NHS England requirements, the evaluation was designed as a longitudinal, mixed-method evaluation, to include the following:

Phase 1:
1. a bespoke patient survey incorporating validated tools
2. a mass matched control analysis.

Phase 2:
1. phased observational interviews with patients
2. weekly diaries, action learning meetings and focus groups with staff and other key stakeholders.

The two phases of the evaluation design are adopting different sampling and recruitment strategies, along with methods, as set out below. Each has its own participant

---

[vi]NIHR portfolio reference number is the same as the IRAS number (IRAS Project ID 208395).

[vii]http://www.goodthingsfoundation.org.

[viii]Stipulated by NHS England.

**Table 3** Technologies available to each cohort

| Cohort | Company | Technology | Description |
|---|---|---|---|
| 1 and 4 | Philips | Motiva* | A telehealth platform that operates through a tablet or television (TV) set top box interface that connects wirelessly to a range of telemonitoring equipment in the home (eg, wireless weighing scales, blood pressure metres, pulse oximeters and thermometers) |
| | MKS Solutions Limited | SpeakSet† | A video calling system that connects a health professional with a patient through the patient's TV in their own home |
| | NHS Stoke-on-Trent Clinical Commissioning Group's Licence | Florence/NHS Simple‡ | A short message communication software providing a reminder/communication text messaging service for patients |
| 1, 2 and 3 | Cambridge Cognition Limited | CANTAB Mobile§ | An assessment tool designed for healthcare professionals to identify the earliest signs of clinically significant memory impairment. The assessment comprises three tests: the Paired Associates Learning test to assess episodic memory; the Geriatric Depression Scale to identify signs of depression; and an activities of daily living questionnaire to assess functionality in daily life. |
| 2 | Intelesant Limited | Intelesant¶ | A mobile app to support daily management of chronic obstructive pulmonary disease through SMS text messaging alerts for coaching, prompts and reminders. It incorporates the individual's action plan in response to changes in symptoms. |
| | Philips | Personal Blood Pressure Cuff** | A blood pressure monitor that works in conjunction with the Philips Suite App to enable a patient to record their blood pressure |
| 2 and 4 | Philips | Health Watch†† | A health watch that tracks heart rate and other cardio condition metrics, and monitors activity, sleeping patterns and nutrition behaviour (not a sports watch). It is best paired with a smartphone running the Philips Suite App. |
| 3 | uMotif Limited | uMotif‡‡ | A health app available as a mobile app, digital tool or wearable device to capture data through a graphical interface and help patients track and understand their health and symptoms. It provides a health report, connects to other wearable devices, sends medication reminders, and keeps track of regular tasks and daily activities. |
| 4 | National Museums Liverpool | House of Memories§§ | A health app that enables people living with dementia and their carers/families to keep a record of objects and experiences from the past. Patients and/or carers can create their own memory tree, memory box or memory timeline. |

*http://www.philips.co.uk/healthcare/solutions/enterprise-telehealth/home-telehealth.
†http://www.speakset.com.
‡http://www.simple.uk.net.
§http://www.cambridgecognition.com.
¶http://www.intelesant.com.
**http://www.philips.co.uk/c-m-hs/health-programs/upper-arm-blood-pressure-monitor.
††http://www.philips.co.uk/c-m-hs/health-programs/health-watch.
‡‡http://www.umotif.com.
§§http://www.liverpoolmuseums.org.uk/learning/projects/house-of-memories/.

information sheets, consent forms and, where applicable, schedules (eg, for the interviews and focus groups). A separate flyer has been designed for patients with dementia, providing information about the study in an informative and accessible manner, drawing on images and large font. In all instances, it will be made clear to

patients that they can be supported by a carer or other family member should they wish. It is anticipated that all participants in cohort 4 (mild dementia) will be accompanied by a carer or other family member.

All materials developed for the evaluation and submitted to ethical review were shared and informed through discussion with patient by experience groups (including those with dementia), a clinical reference group and an evaluation advisory board. Feedback and comment from these three groups were taken into account in finalising the documents submitted for ethics approval.

### The patient survey
#### Recruitment and informed consent
In line with referral to the service, recruitment is being staggered over the programme period. Only patients judged by a clinician to have an acceptable level of cognitive function to give informed consent are invited to take part in the Test Bed. The impact of the programme on each participant is being evaluated over a 6-month period. Final recruitment to the evaluation will thus be at the end of month 15, to allow for the 6-month programme and a 3-month period for final analyses. All patients recruited into the Test Bed and receiving the service are included in the quantitative (service evaluation) stage and are completing the phase 1 consent form. Patients in phase 1 are completing a baseline, mid (12 weeks) and endpoint (24 weeks) survey during the 6-month programme.

#### Methods
The data analysis consists of a statistical evaluation of the effectiveness of the programme and a health economic analysis of its costs and associated cost changes for cohorts 1 and 2. The evaluation is collecting anonymised information for each participant concerning age, gender, ethnicity, living arrangements, education completed, and use of healthcare services in the 4 weeks before completing the questionnaire such as hospital services (inpatient care, outpatient visits and Accident & Emergency visits), primary care services, social care services and medicines. In all cohorts, validated instruments for use with older people to assess health-related quality of life, health and well-being and patient activation are being used (each of which can be completed by proxy if required): Warwick-Edinburgh Mental Well-being Scale, EQ-5D-5L, Patient Activation Measure-13 and the De Jong Gierveld Loneliness Scale. These tools are reapplied at weeks 12 and 24.

The evaluation team are working with the technology innovators[ix] to minimise the demands of the evaluation on patients. Participants are also being offered the option of completing the questionnaires on paper or by telephone.

Drawing on Hospital Episode Statistics data, the primary outcomes for cohorts 1 and 2 will be evaluated against a matched control group (3 controls to 1 Test Bed patient). This will be coordinated in conjunction with the Midlands and Lancashire Commissioning Support Unit.

For cohorts 1 and 2, the cost-effectiveness analysis of the programme will assess the costs per hospital admission avoided. For the Test Bed patients and for the control group, the mean of hospital admissions will be estimated. The primary outcome is the difference between the change in the two groups, which will be estimated using linear regression adjusted for the matching covariates. Ninety-five per cent CIs will be reported for all estimated coefficients. A sensitivity analysis for key parameters will be undertaken.

### The mass control group
The evaluation team will match three control subjects per programme participant for cohorts 1 and 2, the controls being sampled from the same vanguard sites as those on the Test Bed. In terms of contamination between control and Test Bed groups, obtaining control data from the Commissioning Support Unit ensures no controls are also members of the Test Bed. The matching will be performed using the following variables: predicted probability (risk) of hospital admission, LTC, gender and age. The predicted probability of hospital admission for each participant and control will be calculated using the combined predictive model.[12] The matching will be performed using a standard multivariate matching algorithm (Mahalanobis distance matching). The aim of matching on these variables is to obtain a control group which has similar characteristics for these variables compared with the programme participants. This should help minimise any possible confounding effects of these variables.

The statistical analysis is conducted using the statistical software Stata (V.14).

### Phased observational interviews with patients
#### Recruitment and informed consent
Patient participation in phase 2 of the evaluation is optional and participants are asked to indicate their preference for this in the phase 1 consent form. Based on past experience, it is anticipated that more patients will indicate a willingness to take part in the second phase than will be required. Where possible, we are seeking to purposively sample on the basis of chronic health condition(s) (cohorts 1–3), age (55–64, 65–74, 75–84, ≥85), gender, ethnicity, postal address (based on the Office for National Statistics Deprivation Scale), and co-dwelling versus lone dwelling. Ten to fifteen patients are being recruited from each cohort to participate in the indepth qualitative phase of the evaluation (n=40–60).

Those participating in phase 2 are being consented at the time of their first interview. A separate flyer has been designed for patients with dementia to provide information about the Test Bed in an informative and accessible

---

[ix]The term 'innovators' has been coined by NHS England for the purposes of the Test Beds and refers to technology companies deemed to be working at the cutting edge of digital health design and development.

manner. To ensure informed consent and that each participant understands the consent form, the researcher discusses each aspect of the consent form with each participant before it is signed.

Ethical approval was received from the HRA for an alternative consent form with a large font size, and for which clauses on the consent form can be cut into strips, with one clause per strip of paper. This enables the researcher to go through each clause with the participant, focusing on one strip of paper at a time. Once the participant understands the clause clearly, they are asked to initial the clause discussed. This option makes the information in the consent form more manageable for the patient and, prior to ethical approval, was discussed and approved by the Lancaster patient experience group (linked to the NHS). When consenting a person to cohort 4, the researcher talks through the consent form with the patient and in the presence of a family member, friend or carer.

### Methods

Two observational interviews are being undertaken with each participant in their own home over the 6-month period of the programme: (1) in the first month of participating in the programme and (2) during the final month 6. The home-base setting is important as it facilitates a better understanding of participants' health status, health knowledge and activation prior to their participation in the programme, how they engage with the service and use the technology within their own homes at the outset and whether this changes over time. This approach is specifically designed to gain a better understanding of how participants engage with and experience the technology and Fylde Coast Vanguard/Better Care Together services; how this may influence patient activation and self-management of care; any barriers to improving self-care; and whether the programme can increase their sense of empowerment and independence, service satisfaction and overall quality of life.

### Understanding the impact on staff and key stakeholders: diaries, action learning meetings and focus groups

In the early stage of the Test Bed, interviews and focus groups will be conducted with key stakeholders to determine the lessons learned in the process of operationalising the Test Bed. These data will begin to inform and help map out the logic model (see below). Informed consent will be taken for all participants.

All key stakeholders and service providers involved in the delivery of the service are also invited to complete brief weekly diaries and participate in regular 'action learning' meetings, informed by the diary data, as part of a rapid cycle review process. Here, members of the evaluation team meet regularly with key stakeholders involved in the implementation and delivery of the service to reflect on shared learning, agreed action and the impacts of change. This includes technology innovators, members of staff from the Test Bed hubs, voluntary sector providers and

others. Numbers for these meetings are anticipated to be around 20 people, but should numbers increase beyond this we will ask stakeholders to nominate representatives to participate in the meetings and act as two-way conduits for the learning from their particular stakeholder group.

Action learning meetings will take place at 3 monthly intervals and will involve a cycle of discussion, shared learning and reflection and agreed action to be taken forward regarding the operation, delivery and effectiveness of the service. Agreed action from these meetings will build on this cycle of learning and assessment of the service throughout the Test Bed period. To facilitate discussion at these meetings, key stakeholders are asked to complete weekly diaries documenting their experiences of the service, including what works well and less well, and any observations of patients' responses to the service (time spent/support need/technical difficulties). Diaries are brief and simple to complete and are designed to feed into the action research meetings and to fine-tune themes for the focus groups and deliberative panels. The diaries are submitted electronically with participants being encouraged to keep a personal copy as an 'aide memoire' prior to each action learning meeting.

Towards the end of the evaluation, we will also hold four focus groups (n=6–10 participants per focus group) with key stakeholders involved in the delivery of the service. Our objective here is to recruit those directly involved in delivering the service to older participants and those organising and managing it. Recruitment to the focus groups will be split evenly across the two sites. The aim here will be to gain an indepth understanding of how Fylde Coast Vanguard/Better Care Together services with the technology are impacting on working practices. The focus groups will also explore the extent to which the programme may help to increase communication between the care teams, productivity, capacity and coordination of care, and overall work satisfaction.

For ease of reference, a summary of the overall evaluation methods is set out in table 4.

## ETHICS AND DISSEMINATION
### Audio recording

All qualitative data are being audio-recorded using an encrypting digital voice recorder. In the process of transcription, names of participants are anonymised and any identifying features removed or coded. These points are included in the consent forms and are discussed with each participant before the interview/action research meeting/focus group/deliberative panel. All audio files will be deleted from the recorder once data analysis is complete.

### Data analysis

The qualitative data will be analysed thematically with the aid of Atlas.ti qualitative software. Initial identification of emerging themes and initial coding will be undertaken by the field researcher, then shared and agreed during

**Table 4** Summary of evaluation methods being applied to Test Bed participants

| Participants | Survey | Observational interview | Focus groups | Diaries | Action learning meetings | Deliberative panels |
|---|---|---|---|---|---|---|
| Patients (with carers as appropriate) | Yes (3 over 6 months) | Yes (2 over 6 months) | NA | NA | NA | Sample |
| Staff delivering service | NA | NA | Yes | Yes | Yes | Sample |
| Other key stakeholders (technology innovators, voluntary sector and others) | NA | NA | Yes | NA | Yes | Sample |
| Managers/administrators | NA | NA | Yes | NA | NA | Sample |

NA, not applicable.

evaluation team data analysis workshops. This will allow for the verification of the coding framework and enable the evaluation team to develop an analysis of the situation and understand the phenomenon being explored.[13] The major codes emerging from the data will be presented to the deliberative panels for fine-tuning and final verification.

Online supplementary appendix 1 contains information regarding participant payment, data storage, participant benefit, potential risk to participants, withdrawal and researcher risk.

### Deliberative panels

In the final 3 months of the evaluation, we will hold two deliberative panels (one per vanguard site: n=15–20 per panel) with key stakeholders to include older participants; members of the vanguard care teams; key stakeholders from clinical and community settings, and social

care; innovators; and senior managers and commissioners from the Test Bed sites.

Deliberative panels are designed to validate and fine-tune outcomes by producing an informed and collective view resulting from deliberation. They offer a bottom-up approach that can shift evaluation findings into outcomes related to policy, practice and guidance for employers. To achieve this, draft findings emerging from the data analyses will be presented with the aim of drawing on the various expertise of the participants.

In this case, it will help to identify and validate those key components of the programme that will comprise a new model of care that may be scaled up for wider roll-out and will help to finalise the evaluation recommendations. The deliberative panels will also be used to begin preliminary discussions of how this model might form the basis of a joint health and social care strategy for telehealth/

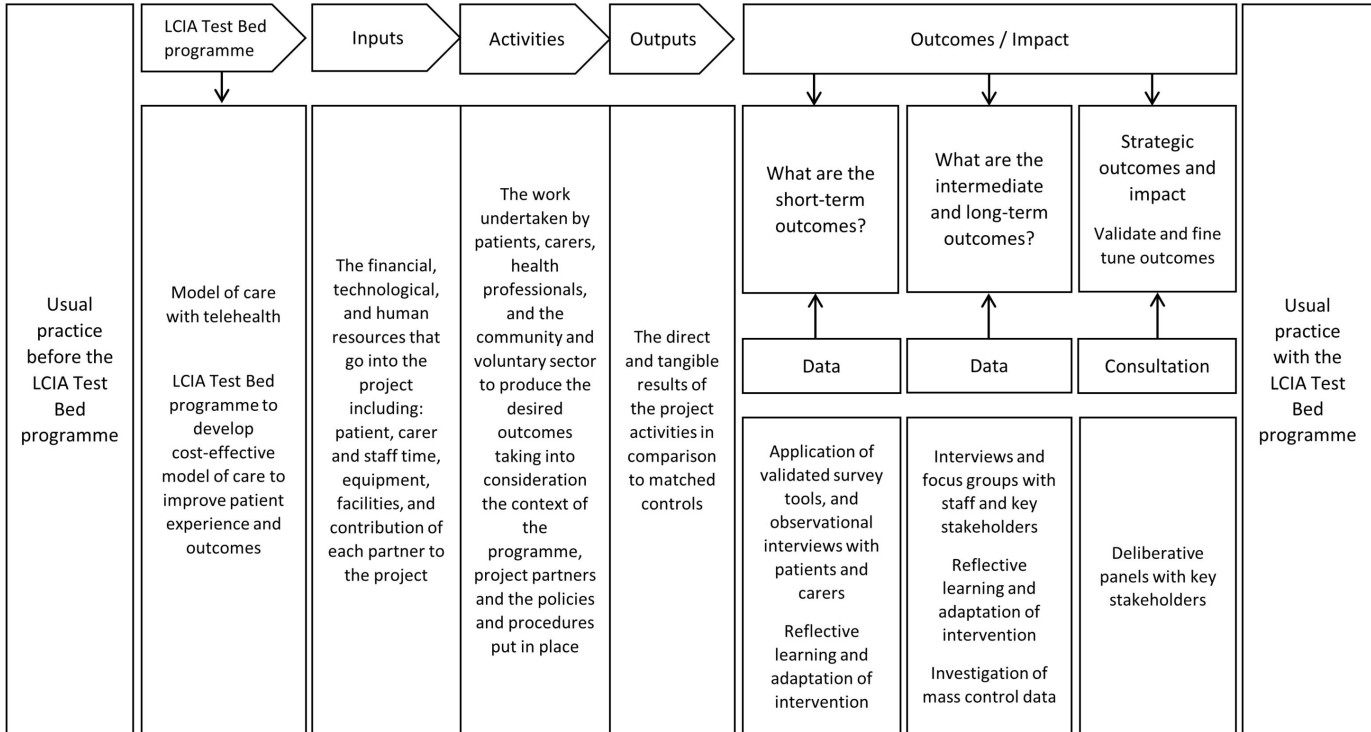

**Figure 1** Logic model for Test Bed evaluation. LCIA, Lancashire and Cumbria Innovation Alliance.

telecare. Finally, the panels will help us to fine-tune the logic model through a process of 'backward mapping' (see below).

### The logic model framework

The evaluation is located within a logic model framework, enabling us to understand the impact of the programme. It also enables us to consider the processes, management and participation that may have implications for the Test Bed's success. In developing the logic model, we are considering the following key elements:

► Activities—the focus of our model is the New Models of Care with Technology programme. The programme of work is considered in its context, including the partners of the project and the policies and procedures that are in place.

► Inputs—including the financial, human, organisational and material resources of the project. This includes the following stakeholders: older adults taking part in the project, Fylde Coast Vanguard and Better Care Together teams, clinicians, GPs, community nurses, other health professionals, and innovator, voluntary and community sector input.

► Outcomes—the desired and actual results of the project and a logic model that will support roll-out and scalability.

The logic model has been designed as a processual model that will be populated using data gathered in the diaries and action research meetings. The elements identified above will be fine-tuned iteratively over the 2 years of the programme and finalised through a process of 'backward mapping' during the final deliberative panels. In this way, the final logic model, designed to assist with wider roll-out of the programme, will be developed by drawing on informed and collective views resulting from ongoing iteration and deliberation involving all key stakeholders. Figure 1 provides an overview of the logic model being used for the Test Bed evaluation.

### Dissemination

The findings of the evaluation will be disseminated in the following ways:

1. a final evaluation report that also contributes to the national evaluation
2. presentations to all key stakeholders within the vanguard sites and NHS England
3. conference papers and posters presented at regional, national and international conferences
4. publication in a range of high-quality, international, peer-reviewed scientific journals
5. publication in a range of journals/publications relevant to service providers
6. dissemination of evaluation highlights through social media.

**Acknowledgements** The authors wish to thank Amanda Thornton, the patient by experience groups, the clinical reference group and the LCIA Test Bed Evaluation Advisory Board for their helpful feedback on the initial design of the protocol materials. In the original bid documentation, Philips contributed to the development of table 2 presented in this paper.

**Contributors** SV is the principal author and wrote the majority of the manuscript. AH, TMP, CMa, JW, MD and CMi are coauthors, and each contributed to the editing and revision of the manuscript through several iterations. CMi is the principal investigator for the evaluation and is the corresponding author.

**Funding** This work is supported by NHS England.

**Competing interests** None declared.

**Patient consent** Not required.

**Ethics approval** The Health Research Authority (IRAS Project ID: 208395) and the Lancaster University Faculty of Health Research Ethics Committee (reference: FHMREC16025).

**Provenance and peer review** Not commissioned; externally peer reviewed.

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
