## [Reviewer comments · BMJ Open]

ARTICLE DETAILS

TITLE (PROVISIONAL)	How effective and cost effective are innovative combinatorial technologies and practices for supporting older people with long-term conditions to remain well in the community? An evaluation protocol for an NHS Test Bed in Northwest England.
AUTHORS	Varey, Sandra; Hernández, Alejandra; Palmer, Tom; Mateus, Ceu; Wilkinson, Joann; Dixon, Mandy; Milligan, Christine

VERSION 1 – REVIEW

REVIEWER	Neil Chadborn University of Nottingham School of Medicine Institute of Mental Health Triumph Road Nottingham NG7 2TU UK
REVIEW RETURNED	10-Jun-2017

GENERAL COMMENTS	The problem and intervention being evaluated is too complex for the quantitative analysis to be able to give a meaningful answer for effectiveness. Some of the details of case-matched comparator sample are not clear enough to give a definitive judgement. However the main problem is of attribution; the 'complex intervention' includes various technologies which may include telehealth, telemedicine, and diagnostics (individually tailored); there is also support from a third sector organisation; aspects of the Vanguard service may also be part of the intervention (although it is unclear if comparators will be selected from the same geographic area). If a reduction in service use is observed, it will be not be possible to attribute this to technology rather than support from third sector organisation or support from MCP/PAC services. It is unclear if the measures of 'patient activation' will have a comparator sample. If not these will be simply before and after measures, which gives weak evidence on which to judge effectiveness.
--

REVIEWER	Dr Judith Carrier Cardiff University S.Wales UK
REVIEW RETURNED	19-Jul-2017

GENERAL COMMENTS	An extremely well written and comprehensive study protocol that
---

	clearly outlines the qualitative, quantitative and economic measures that will be used to evaluate the two-phase approach, which is comprehensively described. Once this study is complete it will provide valuable information related to the impact of health technologies in supporting older people with LTCs, in addition to measuring the effectiveness of these approaches on a range of outcomes. As the intervention consists of a wide range of technologies, with the technologies dependent on several factors, the effectiveness of any one single intervention will not be evident, but as the evidence regarding the effectiveness of health technologies in people with LTCs per se is scarce this study will certainly add to what is currently known, and will support the development of future technological interventions. The mixed methods approach ensures that the perceptions of service users and the workforce regarding the impact of the intervention will be captured, something that is often missed with evaluation studies. My only extremely minor) criticism is the title, after reading the protocol it explains the study very well but is not particular catchy for the reader!
--	---

VERSION 1 – AUTHOR RESPONSE

We are very pleased with Reviewer 1 and 2's feedback, and thank them for their comments. Reviewer 1 asks some questions which we think are adequately addressed in Reviewer 2's feedback. As this is a protocol for a study which is already now underway, we cannot make changes to its design and methodology. In addition to this, the funder, NHS England, has required an inbuilt flexibility to the evaluation as part of its rapid-response design.

We agree that if the aim of the evaluation were to consider the specific efficacy of each technology then the sample size for our quantitative analysis would be too small to consider the specific effectiveness. However, the aim of the Test Bed and our evaluation is to consider the role of combinatorial health technologies as part of a complex intervention, as a whole, in each cohort. This reflects real world practice if the Test Beds are to be adopted more broadly once the programme has finished.

Our comparator control sample will be selected from the same geographic area. The data are being obtained from the same Clinical Commissioning Groups in which the Test Bed is being undertaken.

As suggested, we have changed the title of the paper and have highlighted this change.

We look forward to hearing from you in due course. Should you require any further information, please do not hesitate to contact me as above.

VERSION 2 – REVIEW

REVIEWER	Neil Chadborn University of Nottingham, UK
REVIEW RETURNED	22-Sep-2017
GENERAL COMMENTS	In Ethical approvals, the study is described as a two-phase evaluation, yet phase 2 is described as research (qualitative). Furthermore in Evaluation design, the study is described as having four phases, where phase 2 is quantitative (matched control analysis).

	Some of the details of case-matched comparator sample are not clear enough to give a definitive judgement of quality of the study. Please provide more details of the case-matched comparator sample. Will comparator sample have access to Vanguard services, or 3rd sector services. Will comparator sample have access to already implemented assistive technology, health apps/watches? Will access to these services be treated as confounders and adjusted for? In some parts of the text it appears that Vanguard services and third sector services are part of the intervention; in which case if control participants also received these services, this should be treated as contamination between the study arms. Will the comparator sample be measured on the patient activation scale (PAM)? It would be helpful to have a table or clear statement to concisely state which measures are being applied to which cohort of patients to report outcomes on which of the four objectives of the study? What is the measure of self-care, as opposed to the impact of technology, Vanguard services or 3rd sector services? What is the measure of compliance or engagement with the intervention (in cohorts 2 and 3 respectively)? What is the validated instrument used to measure HRQoL in people with dementia in Cohort 4? How is mild dementia defined in cohort; is there a maximum level of cognitive impairment for recruitment to this group, if so how is this defined? How is informed consent being addressed for patients with cognitive impairment within cohort 4 and other cohorts? Are self-reported measures (eg HRQoL) valid for people with cognitive impairment in cohorts 1-3, will proxy-reporting be included? Which of the above outcomes will be inputs into the health economic analysis? Repeated paragraph on p10: The economic analysis is considering all four patient cohorts. Information on unit costs is being obtained from official sources (e.g. NHS Reference costs and Unit Costs of Health and Social Care, The British National Formulary Tariffs) and Vanguard sites. What is the detail of the model of “New Models of Care with Technology intervention”? If the evaluation is located within a Logic Model framework – is it possible to see a schematic or description of this framework (at least a first draft, if this is iteratively developing)? This may help our understanding of the interaction of the Vanguard services, the 3rd sector services and the new models of care with technology services?
--	--

VERSION 2 – AUTHOR RESPONSE

We thank the reviewer for their detailed comments and have set out below a table that outlines how we have addressed these issues in the revised protocol paper (please note, the version below may

have lost its formatting; please see attached 'Cover letter' for the following table where we have endeavoured to address in detail each aspect of the reviewer feedback - thank you).

Reviewer's comment / request Authors' response

In Ethical approvals, the study is described as a two-phase evaluation, yet phase 2 is described as research (qualitative). We thank the reviewer for highlighting this. For clarification, the word 'research' has been removed and more information has been added to page 6 about the distinction between the two phases and the different ethical approvals received.

Throughout the document, the word 'intervention' has been replaced with 'new service' or 'programme' to avoid confusion (changes highlighted in yellow).

Furthermore in Evaluation design, the study is described as having four phases, where phase 2 is quantitative (matched control analysis).

To avoid confusion, the word 'phase(s)' has been removed and this section has been reframed (see highlighted text on page 10).

Some of the details of case-matched comparator sample are not clear enough to give a definitive judgement of quality of the study. Please provide more details of the case-matched comparator sample. Will comparator sample have access to Vanguard services, or 3rd sector services. Will comparator sample have access to already implemented assistive technology, health apps/watches? Will access to these services be treated as confounders and adjusted for? In some parts of the text it appears that Vanguard services and third sector services are part of the intervention; in which case if control participants also received these services, this should be treated as contamination between the study arms.

The reviewer is right to raise the issue of potential confounding. The comparator sample receiving treatment as usual is a retrospective sample of patients that have been treated in the same sites prior to the implementation of the intervention. Regarding the assistive technologies, most of the technologies available are provided by the healthcare system and very few are available to the general public (e.g. blood pressure cuff, health watch). In the case of the Philips Health Watch, it has been put on sale to the general public after the start of the Test Bed. As such none of the controls would have used the Health Watch because their data is recorded for the 12 months prior to the start of the Test Bed.

Moreover, each Vanguard site operates under a specific model of care and is organised in a different way with different arrangements for the provision of services. It was part of NHS England specifications to include different Vanguard sites with different arrangements concerning the delivery of health care.

In terms of the magnitude of their possible confounding effects, we consider that the covariates which we are using to match Test Bed participants and controls (predicted probability (risk) of hospital admission, long-term condition, sex, and age) are the most important.

For clarification purposes, we have added some sentences about this on page 11 (Mass control group subsection), as highlighted.

Will the comparator sample be measured on the patient activation scale (PAM)?

The comparator sample is only going to be used to assess the effects in inpatient admissions in cohorts 1 and 2. Therefore we will not collect data on the validated instruments for the controls.

For clarification, we have amended the description of the quantitative analysis (Methods subsection) to make this clearer on page 10 -11.

It would be helpful to have a table or clear statement to concisely state which measures are being applied to which cohort of patients to report outcomes on which of the four objectives of the study?

We agree with the reviewer that this needed to be clearer. As noted in the previous comment, on pages 10-11, we have clarified which quantitative analyses will be applied in each cohort. We have also added a sentence on page 8. On page 13, we present a table which summarises which evaluation methods are being applied to the different Test Bed participants.

What is the measure of self-care, as opposed to the impact of technology, Vanguard services or 3rd sector services?

We thank the reviewer for raising this point. Self-care is being assessed in two ways – Data from the validated instruments to assess Health Related Quality of Life and wellbeing (and specifically Patient Activation Measure PAM13) but also through the more in-depth qualitative data. Repeat observational interviews play an important role in understanding how/if the combinatorial technologies have impacted on people's abilities to self-manage their own care, improve their understanding and levels of confidence in managing their condition.

What is the measure of compliance or engagement with the intervention (in cohorts 2 and 3 respectively)?

Compliance or engagement of participants with the use of services and technologies is assessed by participants completing the 24 week intervention.

What is the validated instrument used to measure HRQoL in people with dementia in Cohort 4? The validated instrument to measure HRQoL in Cohort 4 is the EQ-5D-5L. For clarification purposes, the words 'In all cohorts' has been added in the first paragraph on page 11.

How is mild dementia defined in cohort; is there a maximum level of cognitive impairment for recruitment to this group, if so how is this defined? This is an important issue and is addressed on page 7. A patient's ACE III assessment results contribute to the clinical judgement regarding dementia diagnosis, but the categorisation is not based on this alone and is a broader clinical judgement. As highlighted in yellow, some clarification has been added on page 7.

How is informed consent being addressed for patients with cognitive impairment within cohort 4 and other cohorts?

For clarification purposes, information has been inserted on page 12 (highlighted in yellow) to explain how this is being addressed. Also, a sentence has been added on page 10 (Recruitment and informed consent subsection), as highlighted.

Are self-reported measures (eg HRQoL) valid for people with cognitive impairment in cohorts 1-3, will proxy-reporting be included? Only patients judged by a clinician to have an acceptable level of cognitive function to give informed consent are invited to take part in the Test Bed. Therefore the self-reported measures are valid for all the participants and patients self-complete but with the support of a family carer or health professional if requested. As noted in the previous comment, a sentence has been added on page 10 (Recruitment and informed consent subsection), as highlighted.

The validated measures being used can be completed by proxy if required, as stated on page 11.

Which of the above outcomes will be inputs into the health economic analysis? The health economic analysis will consider changes in hospital admissions as a measure of the relevant outcomes of this intervention for patients in cohorts 1 and 2.

For clarification purposes, the words 'for Cohorts 1 and 2' have been added in the Methods subsection in page 10. Moreover, the objective of the quantitative analysis and health economic analysis has been added on page 11, as highlighted.

Repeated paragraph on p10:

The economic analysis is considering all four patient cohorts. Information on unit costs is being obtained from official sources (e.g. NHS Reference costs and Unit Costs of Health and Social Care, The British National Formulary Tariffs) and Vanguard sites. The paragraph has been deleted.

What is the detail of the model of "New Models of Care with Technology intervention"? Further details have been added in page 5 to clarify this and to signpost should the reader wish for further information

If the evaluation is located within a Logic Model framework – is it possible to see a schematic or description of this framework (at least a first draft, if this is iteratively developing)? This may help our understanding of the interaction of the Vanguard services, the 3rd sector services and the new models of care with technology services?

An additional table has been inserted on pages 14-15

VERSION 3 – REVIEW

REVIEWER	Neil Chadborn University of Nottingham UK
REVIEW RETURNED	22-Dec-2017

GENERAL COMMENTS	This is a complex study involving several different vulnerable groups, several different interventions and multiple sources of data (including non-participant data). I have reviewed the protocol with appropriate depth and rigour, following clarifications and removal of inconsistencies within the previous versions of the manuscript. Therefore these are not "new issues". It should be clarified that analysis of service use (primary outcomes 1 and 3) will only apply to cohorts 1 and 2. At the moment statements within the paper are unclear, for example p7 "All patients consenting to take part in the Test Bed and receiving the technology are required to participate in Phase 1..." This is not the case as cohorts 3 and 4 will not take part in component 2 (of Phase 1) which consists of hospital and other service use as well as the matched control and health economic analysis. Therefore primary outcomes 1 and 3 will only be evaluated for cohorts 1 and 2; thus for cohorts 3 and 4, only outcome 2 will be evaluated (of the primary outcomes). Cost effectiveness should be clarified as avoidance of hospital admission, which is presumably interpreted as an acute exacerbation of illness and therefore failure of self-management. Data governance for the matched comparator group is unclear. The manuscript states that comparator data will be drawn from HES data (p11), but presumably to be equivalent to the sample frame, the data should be drawn from the same Combined Predictive Model dataset.
---

	This algorithm is based on “a comprehensive dataset of patient information, including inpatient (IP), outpatient (OP), and accident & emergency (A&E) data from secondary care sources as well as general practice (GP) electronic medical records.” (reference 13) To avoid contamination, i.e. as stated on p11 “to ensure that no controls are also members of the Test Bed” the recruited sample will have to be avoided when extracting the comparator sample. The process of extracting the comparator is not stated, and there is a risk of breaking anonymisation (identifying sensitive information about named patients) either in the recruited sample or the comparator sample (general population) in this process. These risks and governance issues should be clearly stated when evaluating interlinked datasets, to ensure compliance with Caldicott guardianship and Data Protection Act. Due to data being collected and averaged across two distinct Vanguard programmes, it would be advisable to carry out a sensitivity analysis. This may identify benefits which may otherwise be obscured by variance in the data across the two programme areas.
--	--

VERSION 3 – AUTHOR RESPONSE

1. It should be clarified that analysis of service use (primary outcomes 1 and 3) will only apply to cohorts 1 and 2. At the moment statements within the paper are unclear, for example p7 “All patients consenting to take part in the Test Bed and receiving the technology are required to participate in Phase 1...” This is not the case as cohorts 3 and 4 will not take part in component 2 (of Phase 1) which consists of hospital and other service use as well as the matched control and health economic analysis. Therefore primary outcomes 1 and 3 will only be evaluated for cohorts 1 and 2; thus for cohorts 3 and 4, only outcome 2 will be evaluated (of the primary outcomes).

We agree with the reviewer that this needed to be clearer. For clarification, we have added a sentence in page 7 to describe the outcomes that will be measured in each of the cohorts (highlighted text).

2. Cost effectiveness should be clarified as avoidance of hospital admission, which is presumably interpreted as an acute exacerbation of illness and therefore failure of self-management.

The reviewer is right to raise the need to clarify this point. To avoid confusion, on page 11 we have clarified that the cost-effectiveness analysis will assess the costs per hospital admission avoided (strikethrough and highlighted text).

3. Data governance for the matched comparator group is unclear.

The manuscript states that comparator data will be drawn from HES data (p11), but presumably to be equivalent to the sample frame, the data should be drawn from the same Combined Predictive Model dataset. This algorithm is based on “a comprehensive dataset of patient information, including inpatient (IP), outpatient (OP), and accident & emergency (A&E) data from secondary care sources as well as general practice (GP) electronic medical records.” (reference 13) To avoid

We thank the reviewer for these comments regarding data governance.

The NHS number of the Testbed participants is held by the Midlands and Lancashire CSU. The CSU generates a Testbed ID code which is passed to the Testbed evaluation team in order for us to link

the baseline, 3 month, and 6 month questionnaires of each participant. Therefore the evaluation team never knows the NHS number of the participants. Additionally, in any case the Testbed participants were excluded from the control data by the CSU. Therefore, there is no risk of breaking anonymisation.

Contamination, i.e. as stated on p11 “to ensure that no controls are also members of the Test Bed” the recruited sample will have to be avoided when extracting the comparator sample. The process of extracting the comparator is not stated, and there is a risk of breaking anonymisation (identifying sensitive information about named patients) either in the recruited sample or the comparator sample (general population) in this process. These risks and governance issues should be clearly stated when evaluating interlinked datasets, to ensure compliance with Caldicott guardianship and Data Protection Act.

We received approval for our evaluation from the Fylde Coast and North Lancashire CGG Caldicott Guardians in November 2016.

4. Due to data being collected and averaged across two distinct Vanguard programmes, it would be advisable to carry out a sensitivity analysis. This may identify benefits which may otherwise be obscured by variance in the data across the two programme areas.

This is an important issue and is addressed on page 11. For clarification purposes, we have added that we will perform a sensitivity analysis for key parameters (highlighted text).

VERSION 4 – REVIEW

REVIEWER	Neil Chadborn University of Nottingham UK
REVIEW RETURNED	29-Jan-2018
GENERAL COMMENTS	Thanks to the authors for responding to reviewer's concerns. The protocol is now clear on all details sufficient for critique of anticipated empirical data and potential for repeating the study in future.